# Insulin therapy adherence and its associated factors among diabetic patients in a Ghanaian primary care hospital

Israel Abebrese Sefah[1,2]*, Michael Mensah[1], Araba Ata Hutton-Nyameaye[1], Emmanuel Sarkodie[3], Johanna C. Meyer[4], Brian Godman[4,5], Varsha Bangalee[2]

1 University of Health and Allied Sciences, Ho, Ghana, 2 Discipline of Pharmaceutical Sciences, School of Health Sciences, University of KwaZulu-Natal, Durban, South Africa, 3 University Hospital, Kwame Nkrumah University of Science and Technology, Kumasi, Ghana, 4 Department of Public Health Pharmacy and Management, Sefako Makgatho Health Sciences University, Ga-Rankuwa, South Africa, 5 Strathclyde Institute of Pharmacy and Biomedical Science (SIPBS), University of Strathclyde, Glasgow, United Kingdom

* isefah@uhas.edu.gh

**Data Availability Statement:** All relevant data are within the manuscript.

**Funding:** The author(s) received no specific funding for this work.

## Abstract

### Background

Diabetes mellitus (DM) is a global health problem. Adherence to intensive insulin therapy is necessary to achieve better glycemic control in types 1 and 2 DM. This study aimed to evaluate the extent of adherence to insulin therapy, its predictors and to identify barriers to its adherence.

### Method

This was a cross-sectional survey among adult ($\geq$18 years) diabetic patients who are currently using insulin, either alone or in combination with an oral antidiabetic regimen, and seeking primary care at Kwame Nkrumah University of Science and Technology Hospital in Ghana. A total of one hundred and eight-six patients were conveniently sampled, and interviewed. Insulin adherence was determined using the Medication Adherence Reporting Scale-5. Descriptive statistics, a chi-square test of independence, and a multiple logistic regression analysis were performed using STATA version 14 (StataCorp, TX USA).

### Results

The majority of the patients interviewed were over 60 years (40.32%); female (61.83%); married (68.82%); and had completed secondary education (48.39%). 67.20% of the patients were adherent to insulin therapy. Adherence level was associated with age (p = 0.020), marital status (p = 0.001), employment status (p = 0.012), type of DM (p<0.001), regular follow-up (p = 0.007) and comorbidities (p = 0.002) and was only predicted by the type of DM (aOR = 14.82 C.I 1.34–163.50, p-value = 0.028).

**Competing interests:** The authors have declared
no competing interests exist.

## Conclusion

Adherence to insulin therapy among our study population was suboptimal, which is a concern considering the associated increased risk of complications. Adherence assessment and counselling by healthcare professionals to address barriers to poor adherence must be continually undertaken to achieve optimal glycemic control.

## Impact of findings on practice statements

- Continuous adherence assessment and counselling must be offered to all diabetes mellitus patients on insulin therapy as part of their ambulatory care to help improve outcomes.

- Using the Medication Adherence Reporting Scale-5 to determine patient adherence levels is an easy-to-use and an inexpensive method; however, it should be used with caution due to the potential for misclassification.

- Efforts must be made to provide appropriate strategies to deal with barriers to insulin adherence at ambulatory care clinics as part of the individualized comprehensive diabetic care to reduce diabetic complications.

## Introduction

Diabetes is currently one of the leading causes of death and disability globally [1–4]. It was estimated that 537 million people globally had diabetes in 2021 with associated health expenditures of US$966, which is forecast to be over $1054 billion globally by 2045 if not addressed [2, 5]. The increased costs are driven by increasing prevalence rates, projected to rise by 45% to 783 million adults, i.e. 1 in 8 adults, by 2045 (6). Diabetes is a particular concern in low- and middle-income countries (LMICs), which currently accounts for over 75% of the global diabetes population [3, 6]. This may be due to changing lifestyles with reduced physical activity and more sedentary habits along with cultural habits and increasing urbanization [7, 8]. Overall, due to challenges with human resources and financial challenges to procure medicines and equipment, diabetes has a greater impact on morbidity and mortality in sub-Saharan Africa than any other region globally [9].

Currently in Ghana there are approximately 2.4 million people living with diabetes [10]. Whilst the reported prevalence of diabetes at the national level in Ghana have ranged between 2.80% to 3.95% of the population, higher prevalence rates have been reported in different parts of Ghana and different populations [11–14]. At sub-national levels, higher prevalence rates of diabetes have been reported in some regions [15]. For instance, a prevalence rate of 97 25.2% has been reported in the Ashanti region, which is one of the 18 administrative regions in Ghana [16].

Whilst the vast majority of patients with diabetes have type 2 diabetes whose treatment may include insulin, there are an appreciable proportion of patients with type 1 diabetes in Ghana whose treatment is based solely on insulin therapy [17]. However, there are issues of availability and affordability as well as the monitoring equipment among many patients in LMICs, including those in Ghana, impacting on their utilization [17–19]. This is important since achieving optimal glycaemic control using insulin has been associated with a reduction in complications in both type 1 and type 2 diabetes and a reduction in all-cause mortality [20–24].

Despite extensive research and various interventions designed to improve adherence to treatment of chronic conditions, medication non-adherence persists among patients with diabetes, including those prescribed insulin, leading to poor health outcomes [1, 23–26]. Non-adherence is attributed to several complex intertwined factors [24, 27]. These include the complexity of the medication regimens, the social stigma from injecting in public, weight gain, fear of side effects, their costs especially if high co-payments alongside economic difficulties, inconsistent hospital visits, health literacy, and individual health beliefs regarding medication use [17, 20, 24, 26, 28–31].

Currently, standard insulin, including soluble insulin, insulin isophane (NPH) or premixed insulins (30/70), are listed in the Ghana Essential Medicines List (EML) and reimbursed by the National Health Insurance Scheme (NHIS). Long-acting insulin analogues are currently listed in the EML though not reimbursed by the NHIS affecting access among many poor patients [17, 32–34]. As a result, combined with the increasing prevalence of diabetes, there has been growing use of NPH, premixed insulin, and other listed insulins in Ghana in recent years, with very limited use of long-acting insulin analogues due to their high out-of-pocket payment, whose use promotes adherence due to reduce frequency of dosing [35]. There are also concerns that glucometer test strips to monitor HbA1c levels and insulin syringes and needles are currently not reimbursed by NHIS [17, 36]. As a result, this may negatively impact on adherence to insulin therapy among patients with diabetes in Ghana in practice, which may be exacerbated if patients are not active members of NHIS [37]. Typically, in Ghana and across LMICs, diabetes care teams consist of different healthcare professionals who work together to provide comprehensive care to patients with diabetes at ambulatory clinics. Several studies have shown that pharmacists can help enhance adherence to the use of insulin in clinics by providing adherence counseling and regular follow-up, which adherence is potentially enhanced by the use of smartphones [38–40].

There is currently a paucity of research conducted in Ghana to assess adherence to insulin therapy among diabetes patients, with studies to date skewed towards assessing adherence to oral anti-diabetes medicines [30, 41–45]. Consequently, there is a need to address this information gap in patients prescribed insulin in Ghana given rising rates of patients requiring insulin in Ghana and the associated costs to NHIS and others. As a result, this study was designed to evaluate the extent of adherence to insulin therapy among patients with diabetes attending ambulatory care, along with identifying key barriers to adherence to insulin therapy. It is envisaged that the findings will not only be of interest to key stakeholders in Ghana but also to other LMICs faced with similar challenges, with a growing prevalence of patients with diabetes requiring insulin.

## Materials and method

### Study design and setting

A prospective hospital-based cross-sectional survey was conducted among diabetic patients who are currently using insulin as either the only treatment or part of their diabetic regimen, and attending Kwame Nkrumah University of Science and Technology (KNUST) Hospital in Ghana. KNUST Hospital is a government-owned primary care health facility located in the south-east part of Kumasi metropolis in the Ashanti Region of Ghana. The hospital provides medical, surgical, pharmaceutical, diagnostic and specialized services such as diabetes care and asthma care to staff and students at the university as well, as people living in the surrounding community of the university.

## Target population

The study was conducted among adults (18 years and above) attending the ambulatory care diabetic clinic of KNUST. There were largely composed of staff, university students and attendants of the clinic living in communities surrounding the university.

## Sample size and sampling technique

A sample size of 186 was calculated using the Raosoft Inc. online calculator (http://www. raosoft.com/samplesize.html (accessed on the 15th May, 2023), assuming an adherence level of 50% [23] and an average monthly outpatient attendance of 300, at 90% power and 95% confidence interval and additional 10% increase to account for possible non-response or incomplete data. A convenience sampling method was employed in the recruitment of ambulatory diabetes patients on insulin therapy and seeking care at in the primary care facility in KNUST hospital. This was done by recruiting and interviewing all patients who satisfied the inclusion criteria and consented to participate in the study. Inclusion criteria were adult type 1 and type 2 diabetic patients using insulin as part of their diabetes treatment regimen for at least 6 months. Diabetic patients who were not on insulin, newly diagnosed, and hospitalized diabetes patients managed on insulin therapy at the time of the survey, were excluded from this study.

## Data collection

159 Several methods have been developed previously for the assessment of medication adherence among patients with non-communicable disease, including pill counts, electronic monitoring devices and self-reported methods [46]. The latter method is seen as inexpensive, easy to use and practical as it identifies the concerns contributing to non-adherence compared to the 'gold standard' method of direct observation, which has been shown to be impractical, intrusive and resource intensive [47]. The Medication Adherence Reporting Scale (MARS-5) is an example of a reliable self-reported adherence monitoring instrument which has been validated and used for many chronic conditions including diabetes and hypertension [48, 49].

A structured interviewer-administered questionnaire was adapted from data collection tools that we have used in similar previous studies [29–35]. Questionnaires were administered to study participants by two trained pharmacists (IAS & MM) at the diabetes ambulatory clinic at KNUST hospital from 1st June 2023 to 31st August 2023. Other relevant patient data were extracted from patients' medical records. The questionnaire collected socio-demographic characteristics including age, gender and payment type (NHIS or 100% cash payment) as this is a known barrier of accessibility; patient clinical details including the type of DM, duration of DM from the date of diagnosis, duration of insulin therapy, name and type of insulin and documented comorbidities. Another section of the questionnaire determined insulin adherence using the MARS-5 [49]. The MARS-5 measures a patient's medication adherence using five (5) questions that assess patients' negative medication-taking behavior including forgetfulness, independently varying dosages, skipping doses, quitting use of the drug, and taking fewer medicines than instructed. The adherence assessment was conducted by two clinical pharmacists (IAS, MM). The assessment was undertaken using a 5-point Likert scale (1—always, 2—often, 3—sometimes, 4—rarely, and 5—never) [50, 51]. Responses were summed to a total score of 25 and patients with a score of $\geq 21$ were classified as adherent and those with a score of <21 as non-adherent.

## Data analysis

The collected data were checked manually for completeness and accuracy. Data were subsequently entered onto Microsoft Excel® version 2013 and imported into STAT version 14

(StataCorp, TX USA) for analysis. Descriptive statistics were generated to summarize patient socio-demographic and clinical data. A Chi-square test of independence was used to determine the association between the independent variables collected and the study outcome (adherence versus non-adherence assessed using the MARS-5 scale). A multiple logistic regression analysis was subsequently utilized to identify independent predictors of adherence status by estimating the adjusted odd ratios at a significance level of 95% for only variable with chi-square p-values $\leq 0.05$.

### Ethical approval

Ethical approval was secured from the University of Health and Allied Sciences' Research Ethics Committee (UHAS194 REC A.7 [39] 22–23) and administrative approval was obtained from the management of the KNUST Hospital. Written informed consent was secured from each study participant before the interview and personal identifiers were excluded during the data collection to safeguard confidentiality.

## Results

### Socio-demographic characteristics of participants

A total of 186 patients participated in the study with a response rate of 100%. The majority of participants were over 60 years old (40.32%), followed by the 50–60 age group (24.73%); were female (61.83%) and married (68.82%). An appreciable number of participants had completed secondary education (48.39%) and tertiary education (32.26%). A third (36.56%) of the participants were unemployed and 30.11% self-employed. The most common mode of payment for insulin was "cash and carry", i.e. 100% co-payment (55.91%), followed by "insurance" (30.11%). See Table 1 for further details.

### Clinical characteristics of participants

The majority of the participants (85.48%) had type 2 diabetes. Only 43.21% of the participants had a controlled serum glucose (4-7mmol/L). Just under a third of patients (28.49%) had been diagnosed for between 1–5 years, with just under two thirds taking insulin for less than five years (62.90%). Mixtard 30/70 (Premixed insulin of 30% soluble and 70% isophane insulin) was the predominant type of insulin used, accounting for 91.94% of the cases. Most of the participants (76.88%) injected insulin twice daily, and most (83.87%) were taking oral antidiabetic medications in combination with their insulin therapy. Metformin (75.33%) was the most commonly used oral antidiabetic medication. The majority of participants (62.70%) had a glucometer at home for self-monitoring of their insulin. Most participants (90.32%) had regular follow-up appointments, with the majority (75.68%) receiving diabetic education. Most participants (70.81%) had comorbidities, with "hypertension" being the most prevalent (92.48%). The overall rate of adherence to insulin therapy was 67.20% (Table 2). Participants reported various barriers to insulin use including "cost" (34.5%), "side effects" (23.3%) and a combination of "cost and side effects" (23.3%) (Fig 1).

### Association between insulin therapy adherence and patients' characteristics

Adherence to insulin therapy was statistically associated with age (p = 0.020), marital status (p = 0.001), employment status (p = 0.012), type of DM (p<0.001), taking oral antidiabetic medication (p = 0.002), regular follow-up (p = 0.007) and comorbidities (p = 0.002) (Table 3).

Table 1. Socio-demographic characteristics of study participants (n = 186).

| Variables | Frequency (n) |
|---|---|
| Age (in years) | |
| 18–20 | 10 (5.38%) |
| 21–30 | 13 (6.99%) |
| 31–40 | 14 (7.53%) |
| 41–50 | 28 (15.05%) |
| 50–60 | 46 (24.73%) |
| >60 | 75 (40.32%) |
| Sex | |
| Female | 115 (61.83%) |
| Male | 71 (38.17%) |
| Marital status | |
| Married | 128 (68.82%) |
| Divorced | 10 (5.38%) |
| Single | 23 (12.37%) |
| Widowed | 25 (13.44%) |
| Highest educational level | |
| No formal education | 27 (14.52%) |
| Primary education | 9 (4.84%) |
| Secondary education | 90 (48.39%) |
| Tertiary education | 60 (32.26%) |
| Employment status | |
| Employed | 32 (17.20%) |
| Retired | 30 (16.13%) |
| Self-employed | 56 (30.11%) |
| Unemployed | 68 (36.56%) |
| Payment method | |
| Cash and carry | 104 (55.91%) |
| Cash and carry +insurance | 26 (13.98%) |
| Insurance | 56 (30.11%) |

## Independent predictors of adherence to insulin therapy

Patients' adherence to insulin therapy was independently predicted by the type of DM, where patients with type 2 DM were about 15 times more likely to adhere compared to type 1 DM patients (aOR = 14.82 C.I 1.34–163.50, p-value = 0.028), holding the age, marital status, employment status, concomitant use of oral anti-diabetes, presence of co245 morbidities and whether patients regularly visited the clinic constant (Table 4).

## Discussion

Previous studies have demonstrated that good adherence to insulin therapy is an essential requirement for achieving adequate glycemic control and subsequently slowing the progression of microvascular and macrovascular complications, which are mostly associated with DM [2, 28]. The level of adherence to insulin therapy seen in this study was 67.20%, which is seen as suboptimal given the high and growing burden of diabetes in Ghana alongside the growing use of insulin. This is similar to other studies conducted especially in LMICs [52, 53]. Higher adherence rates have been recorded in several other studies elsewhere [54, 55]. Most of the DM patients were taking premixed insulin, with costs lower compared to the longer-acting

**Table 2. Clinical characteristics of participants.**

| Variables | Frequency (n) |
| --- | --- |
| Type of Diabetes mellitus (DM) (n = 186) | |
| Type 1 | 27 (14.52%) |
| Type 2 | 159 (85.48%) |
| Duration of diagnosis (years) (n = 186) | |
| 1–5 | 53 (28.49%) |
| 6–10 | 48 (25.81%) |
| 11–15 | 32 (17.20%) |
| 16–20 | 26 (13.98%) |
| >21 | 27 (14.52%) |
| Duration of insulin therapy (years) (n = 186) | |
| 1–5 | 117 (62.90%) |
| 6–10 | 40 (21.51%) |
| 11–15 | 18 (9.68%) |
| 16–20 | 7 (3.76%) |
| >21 | 4 (2.15%) |
| Name(s) of insulin (n = 186) | |
| Degludec (Long acting insulin analogue) | 2 (1.08%) |
| Isophane | 10 (5.37%) |
| Lantus (Insulin Glargine) | 3 (1.61%) |
| Premixed (30% soluble and 70% isophane) | 171 (91.94%) |
| Frequency of insulin self-injection (n = 186) | |
| Once daily | 43 (23.12%) |
| Twice daily | 143 (76.88%) |
| Oral antidiabetic (n = 186) | |
| Yes | 156 (83.87%) |
| No | 30 (16.13%) |
| Type of oral antidiabetics (n = 156) | |
| Metformin | 116 (75.33%) |
| Metformin and gliclazide | 6 (3.90%) |
| Metformin and pioglitazone | 17 (11.04%) |
| Metformin and vildagliptin | 8 (5.19%) |
| Others | 7 (4.54%) |
| Ownership of glucometer (n = 186) | |
| Yes | 116 (62.70%) |
| No | 69 (37.30%) |
| Regular follow up (n = 186) | |
| Yes | 168 (90.32%) |
| No | 18 (9.68%) |
| Diabetic education (n = 185) | |
| Yes | 140 (75.68%) |
| No | 45 (24.32%) |
| Comorbidities (n = 186) | |
| Yes | 131 (70.81%) |
| No | 54 (29.19%) |
| Type of comorbidity (n = 131) | |
| Hypertension | 123 (92.48%) |
| Hypertension and heart failure | 2 (1.50%) |

(*Continued*)

**Table 2.** (Continued)

| Variables | Frequency (n) |
|---|---|
| Hypertension and Benign prostate hyperplasia | 2 (1.50%) |
| Others | 6 (4.52%) |
| Herbal medicine user (n = 185) | |
| Yes | 4 (2.17%) |
| No | 180 (97.83%) |
| Fasting blood sugar (mmol/L) (n = 162) | |
| 4–7 | 70 (43.21%) |
| >7 | 92 (56.79%) |
| Level of Adherence (n = 186) | |
| Adherent | 125 (67.20%) |
| Non-adherent | 61 (32.80%) |

analogs but have a higher incidence of hypoglycemia compared to the latter [56, 57]. The high risk of hypoglycemia may have contributed among other factors as a barrier to adherence since fear of side effects was cited by patients as a common barrier to adherence. The other important barrier reported by patients was cost as over 50% of them paid for their insulin therapy out of pocket. These barriers must be addressed as part of measures to improve glycemic control among these patients.

The adherence level was found to be associated with patients' age (p = 0.020) with a higher proportion of patients below 31 years exhibiting non-adherence to their insulin therapy. This is consistent with other studies where the busy lifestyles of younger patients, embarrassment of injecting in public, fear of injections, and financial constraints have been reported as important barriers among this young age group, though these influences were not explored in our study [38, 54].

Our study also showed an association between regular visits to healthcare facilities and treatment adherence. This finding is similar to other studies, including a study conducted at

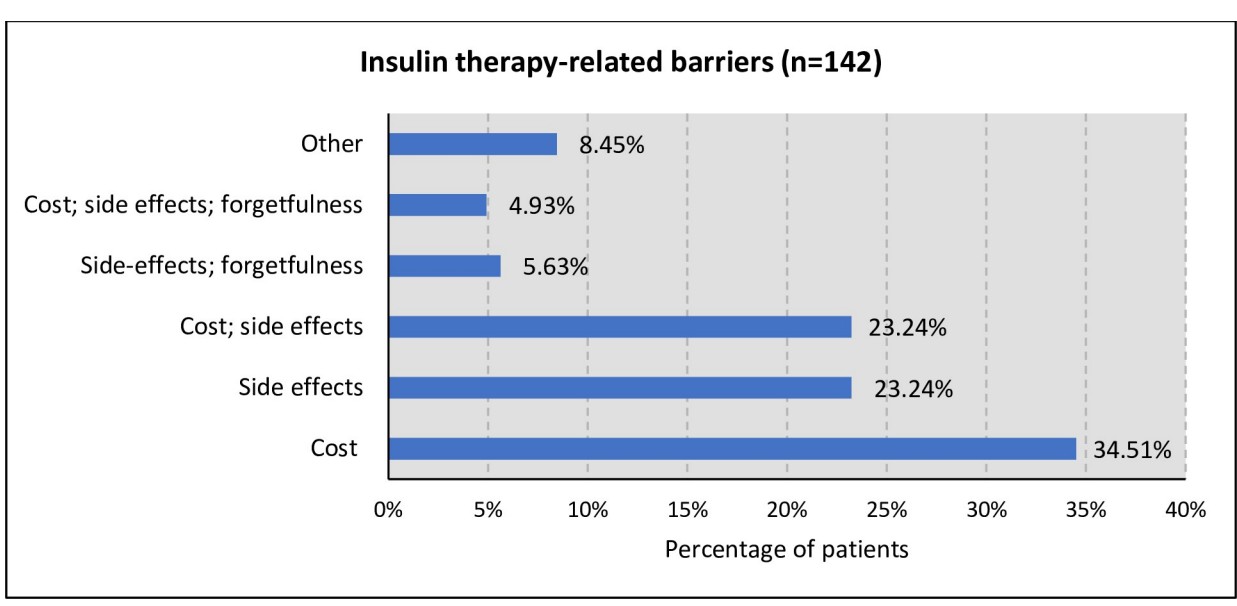

**Fig 1. Common insulin therapy-related barriers patients encountered (n = 142).**

**Table 3. Bivariate analyses of factors associated with adherence to insulin therapy among participants (n = 186).**

| Variables | Total, n (%) | Level of Adherence | | p-value |
|---|---|---|---|---|
| | | Adherent | Non-adherent | |
| Age category (years) | | | | |
| 18–20 | 10 (5.38%) | 3 (30%) | 7 (70%) | **0.020**\* |
| 21–30 | 13 (6.99%) | 5 (38.46%) | 8 (61.54%) | |
| 31–40 | 14 (7.53%) | 9 (64.29%) | 5 (35.71%) | |
| 41–50 | 28 (15.05%) | 20 (71.43%) | 8 (28.57%) | |
| 50–60 | 46 (24.73%) | 32 (69.57%) | 14 (30.43%) | |
| >60 | 75 (40.32%) | 56 (74.67%) | 19 (25.33%) | |
| Gender | | | | |
| Female | 115 (61.83%) | 79 (68.7%) | 36 (31.3%) | 0.581 |
| Male | 71 (38.17%) | 46 (64.79%) | 25 (35.21%) | |
| Marital status | | | | |
| Married | 128 (68.82%) | 92 (71.88%) | 36 (28.13%) | **0.001**\* |
| Divorced | 10 (5.38%) | 7 (70%) | 3 (30%) | |
| Single | 23 (12.37%) | 7 (30.43%) | 16 (69.57%) | |
| Widowed | 25 (13.44%) | 19 (76%) | 6 (24%) | |
| Highest educational level | | | | |
| No formal education | 27 (14.52%) | 21 (77.78%) | 6 (22.22%) | 0.617 |
| Primary education | 9 (4.84%) | 6 (66.67%) | 3 (33.33%) | |
| Secondary education | 90 (48.39%) | 60 (66.67%) | 30(33.33%) | |
| Tertiary education | 60 (32.26%) | 38 (63.33%) | 22 (36.67%) | |
| Employment status | | | | |
| Employed | 32 (17.20%) | 20 (62.5%) | 12 (37.5%) | **0.012**\* |
| Retired | 30 (16.3%) | 24 (80%) | 6 (20%) | |
| Self-employed | 56 (30.11%) | 44 (78.57%) | 12 (21.42%) | |
| Unemployed | 68 (36.56%) | 37 (54.41%) | 31 (45.59%) | |
| Religious status | | | | |
| Christian | 167 (89.78%) | 110 (65.87%) | 57 (34.13%) | 0.250 |
| Muslim | 19 (10.22%) | 15 (78.95%) | 4 (21.05%) | |
| Type of DM | | | | |
| Type 1 | 27 (14.52%) | 8 (29.63%) | 19 (70.37%) | <**0.001**\* |
| Type 2 | 159 (85.48%) | 117 (73.58%) | 42 (26.42%) | |
| Duration of DM diagnosis (years) | | | | |
| 1–5 | 53 (28.49%) | 30 (56.60%) | 23 (43.40%) | 0.344 |
| 6–10 | 48 (25.81%) | 34 (70.833%) | 14 (29.17%) | |
| 11–15 | 32 (17.20%) | 24 (75%) | 8 (25%) | |
| 16–20 | 26 (13.98%) | 17 (65.38%) | 9 (34.62%) | |
| >21 | 27 (14.52%) | 20 (74.07%) | 7 (25.93%) | |
| Duration of insulin therapy (years) | | | | |
| 1–5 | 117 (62.90%) | 82 (70.09%) | 35 (29.9%) | 0.790 |
| 6–10 | 40 (21.51%) | 25 (62.50%) | 15 (37.50%) | |
| 11–15 | 18 (9.68%) | 12 (66.67%) | 6 (33.33%) | |
| 16–20 | 7 (3.76%) | 4 (57.14%) | 3 (42.86%) | |
| >21 | 4 (2.15%) | 2 (50%) | 2 (50%) | |
| Mode of payment for insulin | | | | |

(*Continued*)

**Table 3.** (Continued)

| Variables | Total, n (%) | Level of Adherence | | p-value |
|---|---|---|---|---|
| | | Adherent | Non-adherent | |
| Cash and carry | 104 (55.91%) | 71 (68.27%) | 33 (3.73%) | 0.939 |
| Cash and carry; insurance | 26 (13.98%) | 17 (65.38%) | 9 (34.62%) | |
| Insurance | 56 (30.11%) | 37 (66.07%) | 19 (33.93%) | |
| Frequency of insulin self-injection | | | | |
| Once daily | 43 (23.12%) | 29 (67.44%) | 14 (32.56%) | 0.970 |
| Twice daily | 143 (76.88%) | 96 (67.13%) | 47 (32.87%) | |
| Oral antidiabetic | | | | |
| Yes | 156 (83.87%) | 112 (71.79%) | 44 (28.21%) | **0.002*** |
| No | 30 (16.13%) | 13 (43.33%) | 17 (56.67%) | |
| Having a glucometer at home | | | | |
| Yes | 116 (62.70%) | 81 (69.83%) | 35 (30.17%) | 0.293 |
| No | 69 (37.30%) | 43 (62.32%) | 26 (37.68%) | |
| Regular follow up | | | | |
| Yes | 168 (90.32%) | 118 (70.24%) | 50 (29.76%) | **0.007*** |
| No | 18 (9.68%) | 7 (38.89%) | 11 (611.1%) | |
| Ever had diabetic education | | | | |
| Yes | 140 (75.68%) | 93 (66.43%) | 47 (33.57%) | 0.559 |
| No | 45 (24.32%) | 32 (71.11%) | 13 (28.89%) | |
| Comorbidities | | | | |
| Yes | 131 (70.81%) | 97 (74.05%) | 34 (25.95%) | **0.002*** |
| No | 54 (29.19%) | 27 (50%) | 27 (50%) | |
| Take herbal medication | | | | |
| Yes | 4 (2.17%) | 2 (50%) | 2 (50%) | 0.453 |
| No | 180 (97.83%) | 122 (67.78%) | 58 (32.22%) | |
| Fasting blood sugar (mmol/L) | | | | |
| 4–7 | 70 (43.21%) | 52 (74.29%) | 18 (25.71%) | 0.168 |
| >7 | 92 (56.79%) | 59 (64.13%) | 33 (35.87%) | |

* = Significant at $p < 0.05$

Felege Hiwot Referral Hospital in Ethiopia, which observed that patients who visit healthcare centers regularly were about three times more likely to adhere to insulin therapy compared to those who did not [31, 58, 59]. This could be attributed to the fact that regular follow-up appointments enable patients to maintain consistent communication with their healthcare providers. This ongoing interaction between patients and healthcare providers, including pharmacists, allows for the review, reconciliation and possible adjustment of medication regimens based on patients' evolving health conditions [60, 61]. Pharmacists, as part of the healthcare team, play very critical role by providing adherence counselling regarding patients' disease management and prescribed treatments to help address issues surrounding health literacy regarding their condition [62, 63].

This study also found that individuals with type 1 diabetes exhibited lower adherence to insulin therapy when compared to those with type 2 diabetes (aOR = 14.28, C.I 1.34–163.50, p = 0.028) (Table 4). This finding contrasts with the results of studies conducted in other LMICs [25, 58] where patients with Type 2 diabetes were more likely to be non-adherent to insulin compared to those with Type 1 diabetes. This discrepancy in the findings may be attributed to the fact that the MARS-5 scale assesses independent adjustment of medicine dose

**Table 4. Logistic regression of independent variables which showed a statistically significant association with medication adherence status.**

| Independent variables | aOR | 95% CI | p-value |
|---|---|---|---|
| Age category (years) | | | |
| 18–20Ⓡ | 1.0 | | |
| 21–30 | 0.73 | 0.10–5.32 | 0.753 |
| 31–40 | 0.41 | 0.02–8.57 | 0.562 |
| 41–50 | 0.26 | 0.01–6.04 | 0.398 |
| 50–60 | 0.22 | 0.01–5.34 | 0.352 |
| >60 | 0.23 | 0.01–5.64 | 0.369 |
| Marital status | | | |
| MarriedⓇ | 1.0 | | |
| Divorced | 1.20 | 0.27–5.40 | 0.811 |
| Single | 0.28 | 0.02–3.30 | 0.315 |
| Widowed | 1.84 | 0.60–5.72 | 0.290 |
| Employment status | | | |
| EmployedⓇ | 1.0 | | |
| Retired | 2.55 | 0.62–10.42 | 0.193 |
| Self-employed | 2.31 | 0.82–6.50 | 0.112 |
| Unemployed | 1.21 | 0.41–3.61 | 0.729 |
| Type of DM | | | |
| Type 1Ⓡ | 1.0 | | |
| Type 2 | 14.82 | 1.34–163.50 | **0.028**\* |
| Use of oral antidiabetic | | | |
| Yes | 0.19 | 0.02–2.19 | 0.183 |
| NoⓇ | 1.0 | | |
| Regular visit to a diabetes clinic | | | |
| Yes | 2.18 | 0.63–7.62 | 0.221 |
| NoⓇ | 1.0 | | |
| Presence of Comorbidities | | | |
| Yes | 1.73 | 0.65–4.58 | 0.270 |
| NoⓇ | 1.0 | | |

\* = Significant at P-value <0.05; aOR = Adjusted Odds Ratio; CI = Confidence Interval; Ⓡ = reference variable

as non-adherence to therapy, which may be more common among patients with Type 1 diabetes who frequently monitor their blood sugar levels and independently adjust their insulin doses accordingly. However, we cannot say this with certainty without further studies.

In addition, we found that individuals who were concurrently using oral antidiabetic medication along with insulin demonstrated greater adherence compared to those solely using insulin. This outcome differs from the results obtained in similar studies [52, 53]; but is similar to others which showed low adherence among diabetes patients on insulin alone [31]. This finding could be due to the same reason explained earlier where patients taking only insulin (mainly in type 1 diabetes) are prone to altering their doses which is considered as non-adherence by the MARS-5 scale.

Our study also found that participants who were employed had a better adherence level than their unemployed counterparts. This is similar to other findings where financial constraints were observed to be a determinant of adherence to insulin therapy [54]. Having said this, the mode of payment which is an important economic factors for insulin adherence had

no impact in our study. This needs to be investigated further especially as previous studies undertaken in Ghana and across Africa have called for greater subsidy of the prices of insulin to reduce the barrier of financial constraints to regular insulin usage [17, 19, 30, 32, 35, 36].

Finally, whilst better adherence levels were not found to be associated with good glycemic control (i.e. fasting blood glucose of between 4 to 7 mmol/L) in our study, about three-quarters of participants with good glycemic control were adherent to insulin therapy. The lack of statistical significance of this association could be due to the relatively small sample size used for this study, and we will be looking to investigate this further in the future, considering the urgent need to treat patients with diabetes in Ghana adequately to reduce the level of complications and their associated morbidity, mortality and costs [1]. We are aware of a number of limitations with our study. Firstly, the use of fasting blood sugar levels to measure glycemic control instead of glycated hemoglobin levels may have impacted on the findings. Secondly, the type of adherence assessment tool may also have impacted on the findings, as observed above, though this tool has shown validity and reliability for several adherence studies of chronic diseases. The sample size in this study may have had an impact on the robustness of our study and the statistically significant association of our outcome variable with the many independent variables assessed. Another important limitation of the study is its failure to assess the other component of the comprehensive diabetes care such counseling and patients' education of their disease and medication which could have confounded our findings. Lastly, there may be a limitation of external validity as the demography of the patients who attend the diabetes clinics were mainly university staff and students. However, we believe that the findings from our study will provide useful information to help improve the care of patients using insulin either alone or in combination with other oral therapies in Ghana and other similar settings in LMICs.

## Conclusion

The level of adherence to insulin therapy at KNUST hospital was found to be sub-optimal, though this was more commonly seen among type 1 DM patients. Adherence levels were associated with patients' ages, marital status, type 2 diabetes diagnosis, regular visits to the clinic, concomitant use of oral antidiabetic medication, employment status, and the presence of co-morbidities. Some identified barriers to insulin therapy included cost, side effects, forgetfulness and a combination of cost and side effects, which need to be addressed going forward.

### Recommendations

Adherence assessment and counselling by diabetes care providers on therapy-related barriers including how to deal with side effects and forgetfulness must be continually offered to patients on insulin therapy to secure their greater adherence. This must be part of the individualized and comprehensive diabetic care that targets patient-specific barriers to insulin to achieve normal glycemic control, which will help to reduce diabetic complications occurrence and improve patient's quality of life. Pharmacists can play an active part in this process. Future studies to investigate the effectiveness of different interventions that can improve insulin therapy adherence in Ghana should be undertaken given the increasing burden of diabetes and its complications in the country.

## Author Contributions

**Conceptualization:** Israel Abebrese Sefah, Michael Mensah.

**Data curation:** Israel Abebrese Sefah, Brian Godman, Varsha Bangalee.

**Formal analysis:** Israel Abebrese Sefah, Michael Mensah, Brian Godman, Varsha Bangalee.

**Investigation:** Israel Abebrese Sefah, Michael Mensah, Araba Ata Hutton-Nyameaye.

**Methodology:** Israel Abebrese Sefah, Michael Mensah, Araba Ata Hutton-Nyameaye, Varsha Bangalee.

**Project administration:** Israel Abebrese Sefah, Michael Mensah, Araba Ata Hutton-Nyameaye, Emmanuel Sarkodie.

**Resources:** Israel Abebrese Sefah, Michael Mensah.

**Software:** Israel Abebrese Sefah, Michael Mensah.

**Supervision:** Israel Abebrese Sefah, Araba Ata Hutton-Nyameaye, Emmanuel Sarkodie, Varsha Bangalee.

**Validation:** Israel Abebrese Sefah, Michael Mensah, Araba Ata Hutton-Nyameaye.

**Visualization:** Israel Abebrese Sefah, Michael Mensah, Brian Godman, Varsha Bangalee.

**Writing – original draft:** Israel Abebrese Sefah, Michael Mensah, Araba Ata Hutton-Nyameaye, Emmanuel Sarkodie, Johanna C. Meyer, Brian Godman, Varsha Bangalee.

**Writing – review & editing:** Israel Abebrese Sefah, Michael Mensah, Araba Ata Hutton-Nyameaye, Emmanuel Sarkodie, Johanna C. Meyer, Brian Godman, Varsha Bangalee.

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
