## [Decision Letter · Decision Letter 0]

31 Jul 2024

PONE-D-24-23124Insulin Therapy Adherence and its Associated Factors among Diabetic Patients in a Ghanaian Primary Care HospitalPLOS ONE

Dear Dr. Sefah,

Thank you for submitting your manuscript to PLOS ONE. After careful consideration, we feel that it has merit but does not fully meet PLOS ONE’s publication criteria as it currently stands. Therefore, we invite you to submit a revised version of the manuscript that addresses the points raised during the review process.

We look forward to receiving your revised manuscript.

Kind regards,

Kahsu Gebrekidan, Ph.D.

Academic Editor

PLOS ONE

Journal Requirements:

Reviewers' comments:

Reviewer's Responses to Questions

**Comments to the Author**

1. Is the manuscript technically sound, and do the data support the conclusions?

Reviewer #1: Partly

Reviewer #2: Yes

2. Has the statistical analysis been performed appropriately and rigorously? 

Reviewer #1: Yes

Reviewer #2: Yes

3. Have the authors made all data underlying the findings in their manuscript fully available?

Reviewer #1: Yes

Reviewer #2: Yes

4. Is the manuscript presented in an intelligible fashion and written in standard English?

Reviewer #1: Yes

Reviewer #2: Yes

5. Review Comments to the Author

Reviewer #1: Dear Authors,

Thank you for your submission. Please note the following comments and suggestions

Abstract

- Background - please introduce a sentence to provide an insight into the problem before stating the last sentence on the purpose of the study.

Introduction

L93, it is important to insert the year of reference instead of "currently".

Is there any reason for prevalent in some specific regions etc?.

There is a need to strengthen the novelty of the study.

It is suggested that authors conduct a literature review on insulin therapy adherence. Please insert a section on that.

Method

Does the researchers have an idea of the study total population size before the adopted sample size?.

Discussion

There is a need to explore through the results, the impact of lifestyle and status on the study focus. Also, in the introduction, Authors mentioned the disparate records from the different regions in Ghana, does the study explore this aspect?

More can be achieved from the collected data.

Please create a section for the implication of the study.

Conclusion

Kindly review and enhance this to provide a robust Conclusion to the study.

Please review the manuscript for grammatical errors.

Reviewer #2: - The authors provide a good argument for the survey. The review of the literature is adequate, however I disagree that Insulin therapy is solely responsible for the outcomes in patients with diabetes mellitus (line 106 -108).

- The aim of the study is clear

- The target population is well defined

- The inclusion criteria is well defined

- The authors provide a clear description of the study processes, the data collection instrument and the statistical instruments used in the data analysis.

- The results are presented clearly in accurate tables

- The discussion is adequate but the authors should please look at the reviewer comments made on lines 263, 293-296, 303 - 311. The authors show a clear understanding of the limitations of the study method and processes

- The concluding statement is adequate. The recommendations made are reasonable based on the findings.

- The authors should please make minor corrections to the references high lighted. 10,11 and 34,35.

6. PLOS authors have the option to publish the peer review history of their article (what does this mean?). If published, this will include your full peer review and any attached files.

Reviewer #1: No

Reviewer #2: No

---

## [Author Response · Author response to Decision Letter 0]

1 Aug 2024

REVIEWERS’ COMMENTS AND AUTHORS’ RESPONSES

Insulin Therapy Adherence and its Associated Factors among Diabetic Patients in a Ghanaian Primary Care Hospital

Reviewer #1: Dear Authors,

Thank you for your submission. Please note the following comments and suggestions

Abstract- Background - please introduce a sentence to provide an insight into the problem before stating the last sentence on the purpose of the study.

AUTHOR RESPONSE AND ACTION TAKEN: Thank you for this. We have now added a statement defining the problem to the background section of the abstract

Introduction

L93, it is important to insert the year of reference instead of "currently".

Is there any reason for prevalent in some specific regions etc?.

There is a need to strengthen the novelty of the study.

AUTHOR RESPONSE AND ACTION TAKEN: Thank you for this suggestion. We have now added the year of reference and additional details to strengthen the novelty and relevance of the study.

It is suggested that authors conduct a literature review on insulin therapy adherence. Please insert a section on that.

AUTHOR RESPONSE AND ACTION TAKEN: Thank you for this. We have now added more details to show literature review conducted on the subject matter (line 102 to 106).

Method

Does the researchers have an idea of the study total population size before the adopted sample size?.

AUTHOR RESPONSE AND ACTION TAKEN: We based the calculation of our sample size on the average monthly outpatient attendance of 200 giving us a total of 600 patients to be expected at the clinic over the 3 months’ duration of data collection. We have now updated that section of the method to bring clarity to our estimation (line 152 to 155)

Discussion

There is a need to explore through the results, the impact of lifestyle and status on the study focus. Also, in the introduction, Authors mentioned the disparate records from the different regions in Ghana, does the study explore this aspect?

More can be achieved from the collected data.

AUTHOR RESPONSE AND ACTION TAKEN: Thank you for this comment. Our study was done in just one hospital which provides service to patients living just around the hospital and so we could not explore the different prevalence levels at different regions and settings. We have indicated this as a limitation to our study (line303 to 305).

Please create a section for the implication of the study.

AUTHOR RESPONSE AND ACTION TAKEN: 

Conclusion

Kindly review and enhance this to provide a robust Conclusion to the study.

AUTHOR RESPONSE AND ACTION TAKEN: 

Please review the manuscript for grammatical errors.

AUTHOR RESPONSE AND ACTION TAKEN: Thank you for this. We have now updated the manuscript by correcting identified grammatical errors.

Reviewer #2: - The authors provide a good argument for the survey. The review of the literature is adequate, however I disagree that Insulin therapy is solely responsible for the outcomes in patients with diabetes mellitus (line 106 -108).

- The aim of the study is clear

- The target population is well defined

- The inclusion criteria is well defined

AUTHOR RESPONSE AND ACTION TAKEN: Thank you for your comments.

- The authors provide a clear description of the study processes, the data collection instrument and the statistical instruments used in the data analysis.

AUTHOR RESPONSE AND ACTION TAKEN: Thank you for this.

- The results are presented clearly in accurate tables

AUTHOR RESPONSE AND ACTION TAKEN: We are very grateful for this comment.

- The discussion is adequate but the authors should please look at the reviewer comments made on lines 263, 293-296, 303 - 311. The authors show a clear understanding of the limitations of the study method and processes

AUTHOR RESPONSE AND ACTION TAKEN: Thank you for this. We have now updated the manuscript based on the comments made concerning those portions of the paper to more details on the relevance of our findings and the limitations in our study

- The concluding statement is adequate. The recommendations made are reasonable based on the findings.

AUTHOR RESPONSE AND ACTION TAKEN: We are very grateful for this.

- The authors should please make minor corrections to the references high lighted. 10,11 and 34,35.

AUTHOR RESPONSE AND ACTION TAKEN: We have now corrected the errors in the references identified. Thank you.

---

## [Decision Letter · Decision Letter 1]

23 Sep 2024

PONE-D-24-23124R1Insulin Therapy Adherence and its Associated Factors among Diabetic Patients in a Ghanaian Primary Care HospitalPLOS ONE

Dear Dr. Sefah,

Thank you for submitting your manuscript to PLOS ONE. After careful consideration, we feel that it has merit but does not fully meet PLOS ONE’s publication criteria as it currently stands. Therefore, we invite you to submit a revised version of the manuscript that addresses the points raised during the review process.

We look forward to receiving your revised manuscript.

Kind regards,

Kahsu Gebrekidan, Ph.D.

Academic Editor

PLOS ONE

Journal Requirements:

Reviewers' comments:

Reviewer's Responses to Questions

**Comments to the Author**

1. If the authors have adequately addressed your comments raised in a previous round of review and you feel that this manuscript is now acceptable for publication, you may indicate that here to bypass the “Comments to the Author” section, enter your conflict of interest statement in the “Confidential to Editor” section, and submit your "Accept" recommendation.

Reviewer #2: All comments have been addressed

Reviewer #3: (No Response)

2. Is the manuscript technically sound, and do the data support the conclusions?

Reviewer #2: Yes

Reviewer #3: Yes

3. Has the statistical analysis been performed appropriately and rigorously? 

Reviewer #2: Yes

Reviewer #3: Yes

4. Have the authors made all data underlying the findings in their manuscript fully available?

Reviewer #2: Yes

Reviewer #3: Yes

5. Is the manuscript presented in an intelligible fashion and written in standard English?

Reviewer #2: Yes

Reviewer #3: Yes

6. Review Comments to the Author

Reviewer #2: The revision has been completed to the satisfaction of this reviewer. I have no further comments to add.

Reviewer #3: There were omission in line 51- it should be Kwame Nkrumah University of Technology, this is repeated in line 140. In line 46, the authors should be clear on which patients are place on insulin by stating all types is not clear. the sampling utilised in the study was convenience sampling, author was silent on how this was done at the facility and how many days were used to recruit participants for the study. The abstract of the study needs review especially on all types of diabetes. The inclusion criteria included persons older than 18 years however, in the results section, the age range was between 16-20, author should address the difference in age. It is not clear if all the type 2 diabetes were on the combine medication of insulin and oral glycemic agents.

the references had a different font.

7. PLOS authors have the option to publish the peer review history of their article (what does this mean?). If published, this will include your full peer review and any attached files.

Reviewer #2: No

Reviewer #3: No

---

## [Author Response · Author response to Decision Letter 1]

24 Sep 2024

AUTHORS RESPONSE TO REVIEWER COMMENTS

REVIEWER 2

Comment: The authors have given a clear description of the processes for the sampling and administering of the data collection tool.

Authors response: We are grateful for this comment.

Comment: The data analysis used had adequate statistical instruments to test the associations between dependent and independent variables.

Authors response: We are grateful for this comment.

Comment: The presentation of the results are clear and precise. The tables are accurate

Authors response: Thank you for this comment.

Comment: The authors need to review the document for minor errors such as this - level of adherence OR adherence level...

Authors response: Thank you for this correction. The document has now been updated by removing the error “level of”.

Comment: This argument is unclear as the authors state in the opening statement to the paragraph that the employed were less likely to be non-adherent. The mode of payment may be an entirely different argument. What did the authors mean by 'mode of payment' in the MARS-5 tool that was used?

Authors response: Thank you for comment. The mode of payment was not part of the MARS-5 question but was added to assess how cash payment and/ or insurance has an influence on adherence. We believe that cash payment for insulin and employment status are all economic factors that may influence adherence level. We have added extra details in our manuscript to explain this (Lines 308-309).

Comment: Another limitation of this study is the fact that authors did not look at the subjective aspects of control that may limit the level of control in patients. All the arguments are solely focused on adherence to insulin therapy excluding the other aspects of diabetes care that could serve as confounders to the findings especially relating to control of serum glucose levels.

Authors response: Thank you for this input. We have now included this as an important confounding factor that was not assessed (Line 325-327).

Comment The concluding statement is adequate as it focuses on the findings related to the aim of the study.

Authors response: We are grateful for this comment.

Comment: Please review this reference for the appropriate Vancouver style. It also looks like two references put together in error. 

Authors response: Thank you for this correction. The errors with references 10 and 11 are now corrected (Lines 387-390).

Comments: Authors should please correct the errors in the reference list.

Authors’ response: Thank you for this. The references have been updated to Vancouver style.

Comment Please make the necessary corrections to this reference. Please reference Vancouver style.

Authors response: Thank you for this correction. The errors with reference numbers 34 and 35 are now corrected (Line 450 – 452).

REVIEWER 3

Comment: There were omissions in line 51- it should be Kwame Nkrumah University of Technology, this is repeated in line 140.

Authors response: Thank you for this. Corrections have now been made to this update. (Lines 36 and 127)

Comment: In line 46, the authors should be clear on which patients are place on insulin by stating all types is not clear.

Authors response: Thank you for this. The error is now corrected in line 147.

Comment: The sampling utilized in the study was convenience sampling, author was silent on how this was done at the facility and how many days were used to recruit participants for the study. 

Authors response: We have now on how patients were recruited using the convenience sampling technique (line 146-147). The interview duration is captured in the manuscript at line 164-165

Comment: The abstract of the study needs review especially on all types of diabetes. 

Authors response: Thank you for this correction. We have updated the manuscript (Line 30)

Comment: The inclusion criteria included persons older than 18 years however, in the results section, the age range was between 16-20, author should address the difference in age.

Authors response: Thank you for this. We have now corrected the age range to 18-20 years.

Comment: It is not clear if all the type 2 diabetes were on the combine medication of insulin and oral glycemic agents.

Authors response: This is not so in our study. It is worthy to note that a significant number of the patients were on oral therapy (n=156) which is closer to the number of type 2 DM (n=159) in our study.

Comment: The references had a different font.

Authors response: Thank you for this. We have now formatted the references.

---

## [Decision Letter · Decision Letter 2]

1 Oct 2024

Insulin Therapy Adherence and its Associated Factors among Diabetic Patients in a Ghanaian Primary Care Hospital

PONE-D-24-23124R2

Dear Dr. Israel,

We’re pleased to inform you that your manuscript has been judged scientifically suitable for publication and will be formally accepted for publication once it meets all outstanding technical requirements.

Kind regards,

Kahsu Gebrekidan, Ph.D.

Academic Editor

PLOS ONE

Additional Editor Comments (optional):

Reviewers' comments:

Reviewer's Responses to Questions

**Comments to the Author**

1. If the authors have adequately addressed your comments raised in a previous round of review and you feel that this manuscript is now acceptable for publication, you may indicate that here to bypass the “Comments to the Author” section, enter your conflict of interest statement in the “Confidential to Editor” section, and submit your "Accept" recommendation.

Reviewer #3: All comments have been addressed

2. Is the manuscript technically sound, and do the data support the conclusions?

Reviewer #3: Yes

3. Has the statistical analysis been performed appropriately and rigorously? 

Reviewer #3: Yes

4. Have the authors made all data underlying the findings in their manuscript fully available?

Reviewer #3: Yes

5. Is the manuscript presented in an intelligible fashion and written in standard English?

Reviewer #3: Yes

6. Review Comments to the Author

Reviewer #3: Authors have addressed all the comments raised in the first review and I am satisfied with their responses

7. PLOS authors have the option to publish the peer review history of their article (what does this mean?). If published, this will include your full peer review and any attached files.

Reviewer #3: No

---

## [Editor Report · Acceptance letter]

8 Oct 2024

PONE-D-24-23124R2 

PLOS ONE

Dear Dr. Sefah, 

I'm pleased to inform you that your manuscript has been deemed suitable for publication in PLOS ONE. Congratulations! Your manuscript is now being handed over to our production team.

Kind regards, 

on behalf of

Dr. Kahsu Gebrekidan 

Academic Editor

PLOS ONE